# Lifelong Inverse Reinforcement Learning

**Jorge A. Mendez**, **Shashank Shivkumar**, and **Eric Eaton**
Department of Computer and Information Science
University of Pennsylvania
{mendezme,shashs,eeaton}@seas.upenn.edu

## Abstract

Methods for learning from demonstration (LfD) have shown success in acquiring behavior policies by imitating a user. However, even for a single task, LfD may require numerous demonstrations. For versatile agents that must learn many tasks via demonstration, this process would substantially burden the user if each task were learned in isolation. To address this challenge, we introduce the novel problem of *lifelong learning from demonstration*, which allows the agent to continually build upon knowledge learned from previously demonstrated tasks to accelerate the learning of new tasks, reducing the amount of demonstrations required. As one solution to this problem, we propose the first lifelong learning approach to inverse reinforcement learning, which learns consecutive tasks via demonstration, continually transferring knowledge between tasks to improve performance.

## 1   Introduction

In many applications, such as personal robotics or intelligent virtual assistants, a user may want to teach an agent to perform some sequential decision-making task. Often, the user may be able to demonstrate the appropriate behavior, allowing the agent to learn the customized task through imitation. Research in inverse reinforcement learning (IRL) [29, 1, 43, 21, 31, 28] has shown success with framing the learning from demonstration (LfD) problem as optimizing a utility function from user demonstrations. IRL assumes that the user acts to optimize some reward function in performing the demonstrations, even if they cannot explicitly specify that reward function as in typical reinforcement learning (RL).[1] IRL seeks to recover this reward function from demonstrations, and then use it to train an optimal policy. Learning the reward function instead of merely copying the user's policy provides the agent with a portable representation of the task. Most IRL approaches have focused on an agent learning a single task. However, as AI systems become more versatile, it is increasingly likely that the agent will be expected to learn multiple tasks over its lifetime. If it learned each task in isolation, this process would cause a substantial burden on the user to provide numerous demonstrations.

To address this challenge, we introduce the novel problem of *lifelong learning from demonstration*, in which an agent will face multiple consecutive LfD tasks and must optimize its overall performance. By building upon its knowledge from previous tasks, the agent can reduce the number of user demonstrations needed to learn a new task. As one illustrative example, consider a personal service robot learning to perform household chores from its human owner. Initially, the human might want to teach the robot to load the dishwasher by providing demonstrations of the task. At a later time, the user could teach the robot to set the dining table. These tasks are clearly related since they involve manipulating dinnerware and cutlery, and so we would expect the robot to leverage any relevant knowledge obtained from loading the dishwasher while setting the table for dinner. Additionally, we would hope the robot could improve its understanding of the dishwasher task with any additional

knowledge it gains from setting the dining table. Over the robot's lifetime of many tasks, the ability to share knowledge between demonstrated tasks would substantially accelerate learning.

We frame lifelong LfD as an online multi-task learning problem, enabling the agent to accelerate learning by transferring knowledge among tasks. This transfer can be seen as exploiting the underlying relations among different reward functions (e.g., breaking a wine glass is always undesired). Although lifelong learning has been studied in classification, regression, and RL [10, 34, 4], this is the first study of lifelong learning for IRL. Our framework wraps around existing IRL methods, performing lifelong function approximation of the learned reward functions. As an instantiation of our framework, we propose the *Efficient Lifelong IRL* (ELIRL) algorithm, which adapts Maximum Entropy (MaxEnt) IRL [43] into a lifelong learning setting. We show that ELIRL can successfully transfer knowledge between IRL tasks to improve performance, and this improvement increases as it learns more tasks. It significantly outperforms the base learner, MaxEnt IRL, with little additional cost, and can achieve equivalent or better performance than IRL via Gaussian processes with far less computational cost.

## 2   Related Work

The IRL problem is under-defined, so approaches use different means of identifying which reward function best explains the observed trajectories. Among these, maximum margin IRL methods [29, 1] choose the reward function that most separates the optimal policy and the second-best policy. Variants of these methods have allowed for suboptimal demonstrations [32], non-linear reward functions [35], and game-theoretic learning [37]. Bayesian IRL approaches [31, 30] use prior knowledge to bias the search over reward functions, and can support suboptimal demonstrations [33]. Gradient-based algorithms optimize a loss to learn the reward while, for instance, penalizing deviations from the expert's policy [28]. Maximum entropy models [43, 21, 42] find the most likely reward function given the demonstrations, and produce a policy that matches the user's expected performance without making further assumptions on the preference over trajectories. Other work has avoided learning the reward altogether and focuses instead on modeling the user's policy via classification [27].

Note, however, that all these approaches focus on learning a *single* IRL task, and do not consider sharing knowledge between multiple tasks. Although other work has focused on multi-task IRL, existing methods either assume that the tasks share a state and action space, or scale poorly due to their computational cost; our approach differs in both respects. An early approach to multi-task IRL [12] learned different tasks by sampling from a joint prior on the rewards and policies, assuming that the state-action spaces are shared. Tanwani and Billard [38] studied knowledge transfer for learning from multiple experts, by using previously learned reward functions to bootstrap the search when a new expert demonstrates trajectories. Although efficient, their approach does not optimize performance across all tasks, and only considers learning different experts' approaches to one task.

The notion of transfer in IRL was also studied in an unsupervised setting [2, 11], where each task is assumed to be generated from a set of hidden intentions. These methods cluster an initial batch of tasks, and upon observing each new task, use the clusters to rapidly learn the corresponding reward function. However, they do not address how to update the clusters after observing a new task. Moreover, these methods assume the state-action space is shared across tasks, and, as an inner loop in the optimization, learn a single policy for all tasks. If the space was not shared, the repeated policy learning would become computationally infeasible for numerous tasks. Most recently, transfer in IRL has been studied for solving the one-shot imitation learning problem [13, 17]. In this setting, the agent is tasked with using knowledge from an initial set of tasks to generalize to a new task given a single demonstration of the new task. The main drawback of these methods is that they require a large batch of tasks available at training time, and so cannot handle tasks arriving sequentially.

Our work is most similar to that by Mangin and Oudeyer [25], which poses the multi-task IRL problem as batch dictionary learning of primitive tasks, but appears to be incomplete and unpublished. Finn et al. [16] used IRL as a step for transferring knowledge in a lifelong RL setting, but they do not explore lifelong learning specifically for IRL. In contrast to existing work, our method can handle distinct state-action spaces. It is fully online and computationally efficient, enabling it to rapidly learn the reward function for each new task via transfer and then update a shared knowledge repository. New knowledge is transferred in reverse to improve the reward functions of previous tasks (without retraining on these tasks), thereby optimizing all tasks. We achieve this by adapting ideas from lifelong learning in the supervised setting [34], which we show achieves similar benefits in IRL.

# 3 Inverse Reinforcement Learning

We first describe IRL and the MaxEnt IRL method, before introducing the lifelong IRL problem.

## 3.1 The Inverse RL Problem

A Markov decision process (MDP) is defined as a tuple $\langle \mathcal{S}, \mathcal{A}, T, \mathbf{r}, \gamma \rangle$, where $\mathcal{S}$ is the set of states, $\mathcal{A}$ is the set of actions, the transition function $T : \mathcal{S} \times \mathcal{A} \times \mathcal{S} \mapsto [0, 1]$ gives the probability $P(s_{i+1} \mid s_i, a_i)$ that being in state $s_i$ and taking action $a_i$ will yield a next state $s_{i+1}$, $\mathbf{r} : \mathcal{S} \mapsto \mathbb{R}$ is the reward function[2], and $\gamma \in [0, 1)$ is the discount factor. A policy $\pi : \mathcal{S} \times \mathcal{A} \mapsto [0, 1]$ models the distribution $P(a_i \mid s_i)$ over actions the agent should take in any state. When fully specified, an MDP can be solved via linear or dynamic programming for an optimal policy $\pi^*$ that maximizes the rewards earned by the agent: $\pi^* = \mathrm{argmax}_\pi V^\pi$, with $V^\pi = \mathbb{E}_\pi \left[ \sum_i \gamma^i \mathbf{r}(s_i) \right]$.

In IRL [29], the agent does not know the MDP's reward function, and must infer it from demonstrations $\mathcal{Z} = \{\zeta_1, \ldots, \zeta_n\}$ given by an expert user. Each demonstration $\zeta_j$ is a sequence of state-action pairs $[\mathbf{s}_{0:H}, \mathbf{a}_{0:H}]$ that is assumed to be generated by the user's unknown policy $\hat{\pi}^*$. Once the reward function is learned, the MDP is complete and so can be solved for the optimal policy $\pi^*$.

Given an MDP\$\mathbf{r} = \langle \mathcal{S}, \mathcal{A}, T, \gamma \rangle$ and expert demonstrations $\mathcal{Z}$, the goal of IRL is to estimate the unknown reward function $\mathbf{r}$ of the MDP. Previous work has defined the optimal reward such that the policy enacted by the user be (near-)optimal under the learned reward ($V^{\pi^*} = V^{\hat{\pi}^*}$), while (nearly) all other actions would be suboptimal. This problem is unfortunately ill-posed, since it has numerous solutions, and so it becomes necessary to make additional assumptions in order to find solutions that generalize well. These various assumptions and the strategies to recover the user's policy have been the focus of previous IRL research. We next focus on the MaxEnt approach to the IRL problem.

## 3.2 Maximum Entropy IRL

In the maximum entropy (MaxEnt) algorithm for IRL [43], each state $s_i$ is represented by a feature vector $\mathbf{x}_{s_i} \in \mathbb{R}^d$. Each demonstrated trajectory $\zeta_j$ gives a *feature count* $\mathbf{x}_{\zeta_j} = \sum_{i=0}^H \gamma^i \mathbf{x}_{s_i}$, giving an approximate expected feature count $\tilde{\mathbf{x}} = \frac{1}{n} \sum_j \mathbf{x}_{\zeta_j}$ that must be matched by the agent's policy to satisfy the condition $V^{\pi^*} = V^{\hat{\pi}^*}$. The reward function is represented as a parameterized linear function with weight vector $\boldsymbol{\theta} \in \mathbb{R}^d$ as $\mathbf{r}_{s_i} = \mathbf{r}(\mathbf{x}_{s_i}, \boldsymbol{\theta}) = \boldsymbol{\theta}^\top \mathbf{x}_{s_i}$ and so the cumulative reward of a trajectory $\zeta_j$ is given by $\mathbf{r}_{\zeta_j} = \mathbf{r}(\mathbf{x}_{\zeta_j}, \boldsymbol{\theta}) = \sum_{s_i \in \zeta_j} \gamma^i \boldsymbol{\theta}^\top \mathbf{x}_{s_i} = \boldsymbol{\theta}^\top \mathbf{x}_{\zeta_j}$.

The algorithm deals with the ambiguity of the IRL problem in a probabilistic way, by assuming that the user acts according to a MaxEnt policy. In this setting, the probability of a trajectory is given as: $P(\zeta_j \mid \boldsymbol{\theta}, T) \approx \frac{1}{Z(\boldsymbol{\theta}, T)} \exp(\mathbf{r}_{\zeta_j}) \prod_{(s_i, a_i, s_{i+1}) \in \zeta_j} T(s_{i+1} \mid s_i, a_i)$, where $Z(\boldsymbol{\theta}, T)$ is the partition function, and the approximation comes from assuming that the transition uncertainty has little effect on behavior. This distribution does not prefer any trajectory over another with the same reward, and exponentially prefers trajectories with higher rewards. The IRL problem is then solved by maximizing the likelihood of the observed trajectories $\boldsymbol{\theta}^* = \mathrm{argmax}_{\boldsymbol{\theta}} \log P(\mathcal{Z} \mid \boldsymbol{\theta}) = \mathrm{argmax}_{\boldsymbol{\theta}} \sum_{\zeta_j \in \mathcal{Z}} \log P(\zeta_j \mid \boldsymbol{\theta}, T)$. The gradient of the log-likelihood is the difference between the user's and the agent's feature expectations, which can be expressed in terms of the state visitation frequencies $D_s$: $\tilde{\mathbf{x}} - \sum_{\tilde{\zeta} \in \mathcal{Z}_{MDP}} P(\tilde{\zeta} \mid \boldsymbol{\theta}, T) \mathbf{x}_{\tilde{\zeta}} = \tilde{\mathbf{x}} - \sum_{s \in \mathcal{S}} D_s \mathbf{x}_s$, where $\mathcal{Z}_{MDP}$ is the set of all possible trajectories. The $D_s$ can be computed efficiently via a forward-backward algorithm [43]. The maximum of this concave objective is then achieved when the feature counts match, and so $V^{\pi^*} = V^{\hat{\pi}^*}$.

# 4 The Lifelong Inverse RL Problem

We now introduce the novel problem of lifelong IRL. In contrast to most previous work on IRL, which focuses on single-task learning, this paper focuses on online multi-task IRL. Formally, in the lifelong learning setting, the agent faces a sequence of IRL tasks $\mathcal{T}^{(1)}, \ldots, \mathcal{T}^{(N_{max})}$, each of which is an

MDP\r $\mathcal{T}^{(t)} = \langle \mathcal{S}^{(t)}, \mathcal{A}^{(t)}, T^{(t)}, \gamma^{(t)} \rangle$. The agent will learn tasks consecutively, receiving multiple expert demonstrations for each task before moving on to the next. We assume that *a priori* the agent does not know the total number of tasks $N_{max}$, their distribution, or the order of the tasks.

The agent's goal is to learn a set of reward functions $\mathcal{R} = \{\mathbf{r}(\boldsymbol{\theta}^{(1)}), \ldots, \mathbf{r}(\boldsymbol{\theta}^{(N_{max})})\}$ with a corresponding set of parameters $\Theta = \{\boldsymbol{\theta}^{(1)}, \ldots, \boldsymbol{\theta}^{(N_{max})}\}$. At any time, the agent may be evaluated on any previous task, and so must strive to optimize its performance for all tasks $\mathcal{T}^{(1)}, \ldots, \mathcal{T}^{(N)}$, where $N$ denotes the number of tasks seen so far ($1 \leq N \leq N_{max}$). Intuitively, when the IRL tasks are related, knowledge transfer between their reward functions has the potential to improve the learned reward function for each task and reduce the number of expert demonstrations needed.

After $N$ tasks, the agent must optimize the likelihood of all observed trajectories over those tasks:

$$\max_{\mathbf{r}^{(1)}, \ldots, \mathbf{r}^{(N)}} P\left(\mathbf{r}^{(1)}, \ldots, \mathbf{r}^{(N)}\right) \prod_{t=1}^{N} \left(\prod_{j=1}^{n_t} P\left(\zeta_j \mid \mathbf{r}^{(t)}\right)\right)^{\frac{1}{n_t}} , \qquad (1)$$

where $P(\mathbf{r}^{(1)}, \ldots, \mathbf{r}^{(N)})$ is a reward prior to encourage relationships among the reward functions, and each task is given equal importance by weighting it by the number of associated trajectories $n_t$.

## 5 Lifelong Inverse Reinforcement Learning

The key idea of our framework is to use lifelong function approximation to represent the reward functions for all tasks, enabling continual online transfer between the reward functions with efficient per-task updates. Intuitively, this framework exploits the fact that certain aspects of the reward functions are often shared among different (but related) tasks, such as the negative reward a service robot might receive for dropping objects. We assume the reward functions $\mathbf{r}^{(t)}$ for the different tasks are related via a latent basis of reward components $\mathbf{L}$. These components can be used to reconstruct the true reward functions via a sparse combination of such components with task-specific coefficients $\mathbf{s}^{(t)}$, using $\mathbf{L}$ as a mechanism for transfer that has shown success in previous work [19, 26].

This section develops our framework for lifelong IRL, instantiating it following the MaxEnt approach to yield the ELIRL algorithm. Although we focus on MaxEnt IRL, ELIRL can easily be adapted to other IRL approaches, as shown in Appendix D. We demonstrate the merits of the novel lifelong IRL problem by showing that 1) transfer between IRL tasks can significantly increase their accuracy and 2) this transfer can be achieved by adapting ideas from lifelong learning in supervised settings.

### 5.1 The Efficient Lifelong IRL Algorithm

As described in Section 4, the lifelong IRL agent must optimize its performance over all IRL tasks observed so far. Using the MaxEnt assumption that the reward function $\mathbf{r}_{s_i}^{(t)} = \boldsymbol{\theta}^\top \mathbf{x}_{s_i}^{(t)}$ for each task is linear and parameterized by $\boldsymbol{\theta}^{(t)} \in \mathbb{R}^d$, we can factorize these parameters into a linear combination $\boldsymbol{\theta}^{(t)} = \mathbf{L}\mathbf{s}^{(t)}$ to facilitate transfer between parametric models, following Kumar and Daumé [19] and Maurer et al. [26]. The matrix $\mathbf{L} \in \mathbb{R}^{d \times k}$ represents a set of $k$ latent reward vectors that are shared between all tasks, with sparse task-specific coefficients $\mathbf{s}^{(t)} \in \mathbb{R}^k$ to reconstruct $\boldsymbol{\theta}^{(t)}$.

Using this factorized representation to facilitate transfer between tasks, we place a Laplace prior on the $\mathbf{s}^{(t)}$'s to encourage them to be sparse, and a Gaussian prior on $\mathbf{L}$ to control its complexity, thereby encouraging the reward functions to share structure. This gives rise to the following reward prior:

$$P\left(\mathbf{r}^{(1)}, \ldots, \mathbf{r}^{(N)}\right) = \frac{1}{Z\left(\lambda, \mu\right)} \exp\left(-N\lambda \|\mathbf{L}\|_{\mathsf{F}}^2\right) \prod_{t=1}^{N} \exp\left(-\mu\|\mathbf{s}^{(t)}\|_1\right) , \qquad (2)$$

where $Z(\lambda, \mu)$ is the partition function, which has no effect on the optimization. We can substitute the prior in Equation 2 along with the MaxEnt likelihood into Equation 1. After taking logs and re-arranging terms, this yields the equivalent objective:

$$\min_{\mathbf{L}} \frac{1}{N} \sum_{t=1}^{N} \min_{\mathbf{s}^{(t)}} \left\{ -\frac{1}{n_t} \sum_{\zeta_j^{(t)} \in \mathcal{Z}^{(t)}} \log P\left(\zeta_j^{(t)} \mid \mathbf{L}\mathbf{s}^{(t)}, T^{(t)}\right) + \mu\|\mathbf{s}^{(t)}\|_1 \right\} + \lambda\|\mathbf{L}\|_{\mathsf{F}}^2 . \qquad (3)$$

Note that Equation 3 is separably, but not jointly, convex in $\mathbf{L}$ and the $\mathbf{s}^{(t)}$'s; typical multi-task approaches would optimize similar objectives [19, 26] using alternating optimization.

To enable Equation 3 to be solved online when tasks are observed consecutively, we adapt concepts from the lifelong learning literature. Ruvolo and Eaton [34] approximate a multi-task objective with a similar form to Equation 3 online as a series of efficient online updates. Note, however, that their approach is designed for the supervised setting, using a general-purpose supervised loss function in place of the MaxEnt negative log-likelihood in Equation 3, but with a similar factorization of the learned parametric models. Following their approach but substituting in the IRL loss function, for each new task $t$, we can take a second-order Taylor expansion around the single-task point estimate of $\boldsymbol{\alpha}^{(t)} = \operatorname{argmin}_{\boldsymbol{\alpha}} -\sum_{\zeta_j^{(t)} \in \mathcal{Z}^{(t)}} \log P\big(\zeta_j^{(t)} \mid \boldsymbol{\alpha}, T^{(t)}\big)$, and then simplify to reformulate Equation 3 as

$$\min_{\mathbf{L}} \frac{1}{N} \sum_{t=1}^{N} \min_{\mathbf{s}^{(t)}} \left\{ \left(\boldsymbol{\alpha}^{(t)} - \mathbf{L}\mathbf{s}^{(t)}\right)^{\top} \mathbf{H}^{(t)} \left(\boldsymbol{\alpha}^{(t)} - \mathbf{L}\mathbf{s}^{(t)}\right) + \mu\|\mathbf{s}^{(t)}\|_1 \right\} + \lambda\|\mathbf{L}\|_{\mathsf{F}}^2 \ , \qquad (4)$$

where the Hessian $\mathbf{H}^{(t)}$ of the MaxEnt negative log-likelihood is given by (derivation in Appendix A):

$$\mathbf{H}^{(t)} = \frac{1}{n_t} \nabla_{\boldsymbol{\theta},\boldsymbol{\theta}}^2 \mathcal{L}\Big(\mathbf{r}\big(\mathbf{L}\mathbf{s}^{(t)}\big), \mathcal{Z}^{(t)}\Big) = \left(-\sum_{\tilde{\zeta} \in \mathcal{Z}_{MDP}} \mathbf{x}_{\tilde{\zeta}} P(\tilde{\zeta}|\boldsymbol{\theta})\right)\left(\sum_{\tilde{\zeta} \in \mathcal{Z}_{MDP}} \mathbf{x}_{\tilde{\zeta}}^{\top} P(\tilde{\zeta}|\boldsymbol{\theta})\right) + \sum_{\tilde{\zeta} \in \mathcal{Z}_{MDP}} \mathbf{x}_{\tilde{\zeta}} \mathbf{x}_{\tilde{\zeta}}^{\top} P(\tilde{\zeta}|\boldsymbol{\theta}) \ . \ (5)$$

Since $\mathbf{H}^{(t)}$ is non-linear in the feature counts, we cannot make use of the state visitation frequencies obtained for the MaxEnt gradient in the lifelong learning setting. This creates the need for obtaining a sample-based approximation. We first solve the MDP for an optimal policy $\pi^{\boldsymbol{\alpha}^{(t)}}$ from the parameterized reward learned by single-task MaxEnt. We compute the feature counts for a fixed number of finite horizon paths by following the stochastic policy $\pi^{\boldsymbol{\alpha}^{(t)}}$. We then obtain the sample covariance of the feature counts of the paths as an approximation of the true covariance in Equation 5.

Given each new consecutive task $t$, we first estimate $\boldsymbol{\alpha}^{(t)}$ as described above. Then, Equation 4 can be approximated online as a series of efficient update equations [34]:

$$\mathbf{s}^{(t)} \leftarrow \operatorname{argmin}_{\mathbf{s}} \ell\Big(\mathbf{L}_N, \mathbf{s}, \boldsymbol{\alpha}^{(t)}, \mathbf{H}^{(t)}\Big) \quad \mathbf{L}_{N+1} \leftarrow \operatorname{argmin}_{\mathbf{L}} \lambda\|\mathbf{L}\|_{\mathsf{F}}^2 + \frac{1}{N}\sum_{t=1}^{N} \ell\Big(\mathbf{L}, \mathbf{s}^{(t)}, \boldsymbol{\alpha}^{(t)}, \mathbf{H}^{(t)}\Big) \ , \ (6)$$

where $\ell(\mathbf{L}, \mathbf{s}, \boldsymbol{\alpha}, \mathbf{H}) = \mu\|\mathbf{s}\|_1 + (\boldsymbol{\alpha} - \mathbf{L}\mathbf{s})^{\top}\mathbf{H}(\boldsymbol{\alpha} - \mathbf{L}\mathbf{s})$, and $\mathbf{L}$ can be built incrementally in practice (see [34] for details). Critically, this online approximation removes the dependence of Equation 3 on the numbers of training samples and tasks, making it scalable for lifelong learning, and provides guarantees on its convergence with equivalent performance to the full multi-task objective [34]. Note that the $\mathbf{s}^{(t)}$ coefficients are only updated while training on task $t$ and otherwise remain fixed.

This process yields the estimated reward function as $\mathbf{r}_{s_i}^{(t)} = \mathbf{L}\mathbf{s}^{(t)}\mathbf{x}_{s_i}$. We can then solve the now-complete MDP for the optimal policy using standard RL. The complete ELIRL algorithm is given as Algorithm 1. ELIRL can either support a common feature space across tasks, or can support different feature spaces across tasks by making use of prior work in autonomous cross-domain transfer [3], as shown in Appendix C.

---

**Algorithm 1** ELIRL $(k, \lambda, \mu)$

---

$\mathbf{L} \leftarrow \text{RandomMatrix}_{d,k}$
**while** some task $\mathcal{T}^{(t)}$ is available **do**
$\quad \mathcal{Z}^{(t)} \leftarrow \text{getExampleTrajectories}(\mathcal{T}^{(t)})$
$\quad \boldsymbol{\alpha}^{(t)}, \mathbf{H}^{(t)} \leftarrow \text{inverseReinforcementLearner}(\mathcal{Z}^{(t)})$
$\quad \mathbf{s}^{(t)} \leftarrow \operatorname{argmin}_{\mathbf{s}}(\boldsymbol{\alpha}^{(t)} - \mathbf{L}\mathbf{s})^{\top}\mathbf{H}^{(t)}(\boldsymbol{\alpha}^{(t)} - \mathbf{L}\mathbf{s}) + \mu\|\mathbf{s}\|_1$
$\quad \mathbf{L} \leftarrow \text{updateL}(\mathbf{L}, \mathbf{s}^{(t)}, \boldsymbol{\alpha}^{(t)}, \mathbf{H}^{(t)}, \lambda)$
**end while**

---

## 5.2 Improving Performance on Earlier Tasks

As ELIRL is trained over multiple IRL tasks, it gradually refines the shared knowledge in $\mathbf{L}$. Since each reward function's parameters are modeled as $\boldsymbol{\theta}^{(t)} = \mathbf{L}\mathbf{s}^{(t)}$, subsequent changes to $\mathbf{L}$ after training on task $t$ can affect $\boldsymbol{\theta}^{(t)}$. Typically, this process improves performance in lifelong learning [34], but it might occasionally decrease performance through negative transfer, due to the ELIRL

simplifications restricting that $\mathbf{s}^{(t)}$ is fixed except when training on task $t$. To prevent this problem, we introduce a novel technique. Whenever ELIRL is tested on a task $t$, it can either directly use the $\boldsymbol{\theta}^{(t)}$ vector obtained from $\mathbf{Ls}^{(t)}$, or optionally repeat the optimization step for $\mathbf{s}^{(t)}$ in Equation 6 to account for potential major changes in the $\mathbf{L}$ matrix since the last update to $\mathbf{s}^{(t)}$. This latter optional step only involves running an instance of the LASSO, which is highly efficient. Critically, it does not require either re-running MaxEnt or recomputing the Hessian, since the optimization is always done around the optimal single-task parameters, $\boldsymbol{\alpha}^{(t)}$. Consequently, ELIRL can pay a small cost to do this optimization when it is faced with performing on a previous task, but it gains potentially improved performance on that task by benefiting from up-to-date knowledge in $\mathbf{L}$, as shown in our results.

## 5.3 Computational Complexity

The addition of a new task to ELIRL requires an initial run of single-task MaxEnt to obtain $\boldsymbol{\alpha}^{(t)}$, which we assume to be of order $\mathcal{O}(i\xi(d, |\mathcal{A}|, |\mathcal{S}|))$, where $i$ is the number of iterations required for MaxEnt to converge. The next step is computing the Hessian, which costs $\mathcal{O}(MH + Md^2)$, where $M$ is the number of trajectories sampled for the approximation and $H$ is their horizon. Finally, the complexity of the update steps for $\mathbf{L}$ and $\mathbf{s}^{(t)}$ is $\mathcal{O}(k^2d^3)$ [34]. This yields a total per-task cost of $\mathcal{O}(i\xi(d, |\mathcal{A}|, |\mathcal{S}|) + MH + Md^2 + k^2d^3)$ for ELIRL. The optional step of re-updating $\mathbf{s}^{(t)}$ when needing to perform on task $t$ would incur a computational cost of $\mathcal{O}(d^3 + kd^2 + dk^2)$ for constructing the target of the optimization and running LASSO [34].

Notably, there is no dependence on the number of tasks $N$, which is precisely what makes ELIRL suitable for lifelong learning. Since IRL in general requires finding the optimal policy for different choices of the reward function as an inner loop in the optimization, the additional dependence on $N$ would make any IRL method intractable in a lifelong setting. Moreover, the only step that depends on the size of the state and action spaces is single-task MaxEnt. Thus, for high-dimensional tasks (e.g., robotics tasks), replacing the base learner would allow our algorithm to scale gracefully.

## 5.4 Theoretical Convergence Guarantees

ELIRL inherits the theoretical guarantees showed by Ruvolo and Eaton [34]. Specifically, the optimization is guaranteed to converge to a local optimum of the approximate cost function in Equation 4 as the number of tasks grows large. Intuitively, the quality of this approximation depends on how much the factored representation $\boldsymbol{\theta}^{(t)} = \mathbf{Ls}^{(t)}$ deviates from $\boldsymbol{\alpha}^{(t)}$, which in turn depends on how well this representation can capture the task relatedness. However, we emphasize that this approximation is what allows the method to solve the multi-task learning problem online, and it has been shown empirically in the contexts of supervised learning [34] and RL [4] that this approximate solution can achieve equivalent performance to exact multi-task learning in a variety of problems.

# 6 Experimental Results

We evaluated ELIRL on two environments, chosen to allow us to create arbitrarily many tasks with distinct reward functions. This also gives us known rewards as ground truth. No previous multi-task IRL method was tested on such a large task set, nor on tasks with varying state spaces as we do.

**Objectworld:** Similar to the environment presented by Levine et al. [21], Objectworld is a $32 \times 32$ grid populated by colored objects in random cells. Each object has one of five outer colors and one of two inner colors, and induces a constant reward on its surrounding $5 \times 5$ grid. We generated 100 tasks by randomly choosing 2–4 outer colors, and assigning to each a reward sampled uniformly from $[-10, 5]$; the inner colors are distractor features. The agent's goal is then to move toward objects with "good" (positive) colors and away from objects with "bad" (negative) colors. Ideally, each column of $\mathbf{L}$ would learn the impact field around one color, and the $\mathbf{s}^{(t)}$'s would encode how good or bad each color is in each task. There are $d = 31(5 + 2)$ features, representing the distance to the nearest object with each outer and inner color, discretized as binary indicators of whether the distance is less than 1–31. The agent can choose to move along the four cardinal directions or stay in place.

**Highway:** Highway simulations have been used to test various IRL methods [1, 21]. We simulate the behavior of 100 different drivers on a three-lane highway in which they can drive at four speeds. Each driver prefers either the left or the right lane, and either the second or fourth speed. Each driver's

weight for those two factors is sampled uniformly from $[0, 5]$. Intuitively, each column of **L** should learn a speed or lane, and the $\mathbf{s}^{(t)}$'s should encode the drivers' preferences over them. There are $d = 4 + 3 + 64$ features, representing the current speed and lane, and the distances to the nearest cars in each lane in front and back, discretized in the same manner as Objectworld. Each time step, drivers can choose to move left or right, speed up or slow down, or maintain their current speed and lane.

In both environments, the agent's chosen action has a $70\%$ probability of success and a $30\%$ probability of a random outcome. The reward is discounted with each time step by a factor of $\gamma = 0.9$.

## 6.1 Evaluation Procedure

For each task, we created an instance of the MDP by placing the objects in random locations. We solved the MDP for the true optimal policy, and generated simulated user trajectories following this policy. Then, we gave the IRL algorithms the MDP\\**r** and the trajectories to estimate the reward **r**. We compared the learned reward function with the true reward function by standardizing both and computing the $\ell_2$-norm of their difference. Then, we trained a policy using the learned reward function, and compared its expected return to that obtained by a policy trained using the true reward.

We tested ELIRL using **L** trained on various subsets of tasks, ranging from 10 to 100 tasks. At each testing step, we evaluated performance of *all* 100 tasks; this includes as a subset evaluating all previously observed tasks, but it is significantly more difficult because the latent basis **L**, which is trained only on the initial tasks, must generalize to future tasks. The single-task learners were trained on all tasks, and we measured their average performance across all tasks. All learners were given $n_t = 32$ trajectories for Objectworld and $n_t = 256$ trajectories for Highway, all of length $H = 16$. We chose the size $k$ of **L** via domain knowledge, and initialized **L** sequentially with the $\boldsymbol{\alpha}^{(t)}$'s of the first $k$ tasks. We measured performance on a new random instance of the MDP for each task, so as not to conflate overfitting the training environment with high performance. Results were averaged over 20 trials, each using a random task ordering.

We compared ELIRL with both the original (ELIRL) and re-optimized (ELIRLre) $\mathbf{s}^{(t)}$ vectors to MaxEnt IRL (the base learner) and GPIRL [21] (a strong single-task baseline). None of the existing multi-task IRL methods were suitable for this experimental setting—other methods assume a shared state space and are prohibitively expensive for more than a few tasks [12, 2, 11], or only learn different experts' approaches to a single task [38]. Appendix B includes a comparison to MTMLIRL [2] on a simplified version of Objectworld, since MTMLIRL was unable to handle the full version.

Figure 1: Average reward and value difference in the lifelong setting. Reward difference measures the error between learned and true reward. Value difference compares expected return from the policy trained on the learned reward and the policy trained on the true reward. The whiskers denote std. error. ELIRL improves as the number of tasks increases, achieving better performance than its base learner, MaxEnt IRL. Using re-optimization after learning all tasks allows earlier tasks to benefit from the latest knowledge, increasing ELIRL's performance above GPIRL. (Best viewed in color.)

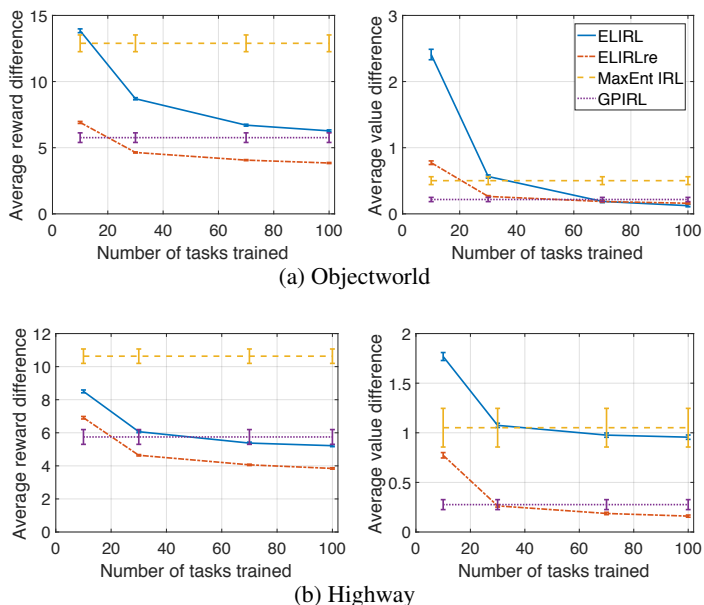

(a) Objectworld

(b) Highway

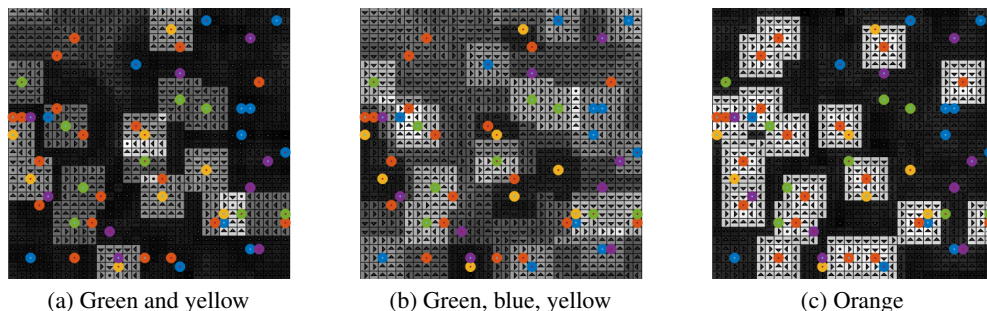

| (a) Green and yellow | (b) Green, blue, yellow | (c) Orange |

Figure 2: Example latent reward functions from Objectworld learned by ELIRL. Each column of **L** can be visualized as a reward function, and captures a reusable chunk of knowledge. The grayscale values show the learned reward and the arrows show the corresponding optimal policy. Each latent component has specialized to focus on objects of particular colors, as labeled. (Best viewed in color.)

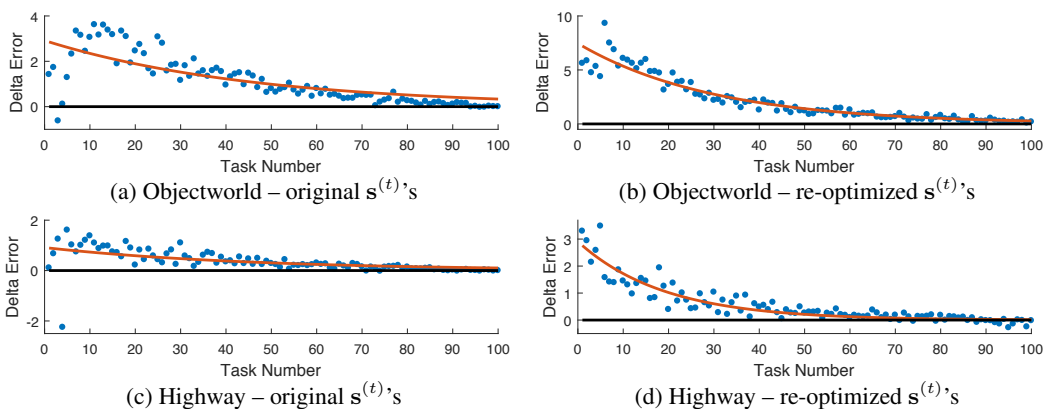

(a) Objectworld – original $\mathbf{s}^{(t)}$'s

(b) Objectworld – re-optimized $\mathbf{s}^{(t)}$'s

(c) Highway – original $\mathbf{s}^{(t)}$'s

(d) Highway – re-optimized $\mathbf{s}^{(t)}$'s

Figure 3: Reverse transfer. Difference in error in the learned reward between when a task was first trained and after the full model had been trained, as a function of task order. Positive change in errors indicates positive transfer; negative change indicates interference from negative transfer. Note that the re-optimization has both decreased negative transfer on the earliest tasks, and also significantly increased the magnitude of positive reverse transfer. Red curves show the best exponential curve.

## 6.2 Results

Figure 1 shows the advantage of sharing knowledge among IRL tasks. ELIRL learned the reward functions more accurately than its base learner, MaxEnt IRL, after sufficient tasks were used to train the knowledge base **L**. This directly translated to increased performance of the policy trained using the learned reward function. Moreover, the $\mathbf{s}^{(t)}$ re-optimization (Section 5.2) allowed ELIRLre to outperform GPIRL, by making use of the most updated knowledge.

As shown in Table 1, ELIRL requires little extra training time versus MaxEnt IRL, even with the optional $\mathbf{s}^{(t)}$ re-optimization, and runs significantly faster than GPIRL. The re-optimization's additional time is nearly imperceptible. This signifies a clear advantage for ELIRL when learning multiple tasks in real-time.

|  | Objectworld (sec) | Highway (sec) |
|---|---|---|
| ELIRL | $17.055 \pm 0.091$ | $21.438 \pm 0.173$ |
| ELIRLre | $17.068 \pm 0.091$ | $21.440 \pm 0.173$ |
| MaxEnt IRL | $16.572 \pm 0.407$ | $18.283 \pm 0.775$ |
| GPIRL | $1008.181 \pm 67.261$ | $392.117 \pm 18.484$ |

Table 1: The average learning time per task. The standard error is reported after the $\pm$.

In order to analyze how ELIRL captures the latent structure underlying the tasks, we created new instances of Objectworld and used a single learned latent component as the reward of each new MDP (i.e., a column of **L**, which can be treated as a latent reward function factor). Figure 2 shows example

Figure 4: Results for extensions of ELIRL. Whiskers denote standard errors. (a) Reward difference (lower is better) between MaxEnt, in-domain ELIRL, and cross-domain ELIRL. Transferring knowledge across domains improved the accuracy of the learned reward. (b) Value difference (lower is better) obtained by ELIRL and AME-IRL on the planar navigation environment. ELIRL improves the performance of AME-IRL, and this improvement increases as ELIRL observes more tasks.

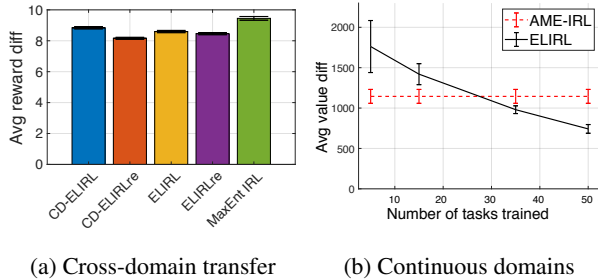

(a) Cross-domain transfer    (b) Continuous domains

latent components learned by the algorithm, revealing that each latent component represents the $5 \times 5$ grid around a particular color or small subset of the colors.

We also examined how performance on the earliest tasks changed during the lifelong learning process. Recall that as ELIRL learns new tasks, the shared knowledge in $\mathbf{L}$ continually changes. Consequently, the modeled reward functions for all tasks continue to be refined automatically over time, without retraining on the tasks. To measure this effect of "reverse transfer" [34], we compared the performance on each task when it was first encountered to its performance after learning all tasks, averaged over 20 random task orders. Figure 3 reveals that ELIRL improves previous tasks' performance as $\bar{\mathbf{L}}$ is refined, achieving reverse transfer in IRL. Reverse transfer was further improved by the $\mathbf{s}^{(t)}$ re-optimization.

### 6.3 ELIRL Extensions to Cross-Domain Transfer and Continuous State-Action Spaces

We performed additional experiments to show how simple extensions to ELIRL can transfer knowledge across tasks with different feature spaces and with continuous state-action spaces.

ELIRL can support transfer across task domains with different feature spaces by adapting prior work in cross-domain transfer [3]; details of this extension are given in Appendix C. To evaluate cross-domain transfer, we constructed 40 Objectworld domains with different feature spaces by varying the grid sizes from 5 to 24 and letting the number of outer colors be either 3 or 5. We created 10 tasks per domain, and provided the agents with 16 demonstrations per task, with lengths varying according to the number of cells in each domain. We compared MaxEnt IRL, in-domain ELIRL with the original (ELIRL) and re-optimized (ELIRLre) $\mathbf{s}^{(t)}$'s, and cross-domain ELIRL with the original (CD-ELIRL) and reoptimized (CD-ELIRLre) $\mathbf{s}^{(t)}$'s, averaged over 10 random task orderings. Figure 4a shows how cross-domain transfer improved the performance of an agent trained only on tasks within each domain. Notice how the $\mathbf{s}^{(t)}$ re-optimization compensates for the major changes in the shared knowledge that occur when the agent encounters tasks from different domains.

We also explored an extension of ELIRL to continuous state spaces, as detailed in Appendix D. To evaluate this extension, we used a continuous planar navigation task similar to that presented by Levine and Koltun [20]. Analogous to Objectworld, this continuous environment contains randomly distributed objects that have associated rewards (sampled randomly), and each object has an area of influence defined by a radial basis function. Figure 4b shows the performance of ELIRL on 50 continuous navigation tasks averaged over 20 different task orderings, compared against the average performance of the single-task AME-IRL algorithm [20] across all tasks. These results show that ELIRL is able to achieve better performance in the continuous space than the single-task learner, once a sufficient number of tasks has been observed.

## 7 Conclusion

We introduced the novel problem of lifelong IRL, and presented a general framework that is capable of sharing learned knowledge about the reward functions between IRL tasks. We derived an algorithm for lifelong MaxEnt IRL, and showed how it can be easily extended to handle different single-task IRL methods and diverse task domains. In future work, we intend to study how more powerful base learners can be used for the learning of more complex tasks, potentially from human demonstrations.

## Acknowledgements

This research was partly supported by AFRL grant #FA8750-16-1-0109 and DARPA agreement #FA8750-18-2-0117. We would like to thank the anonymous reviewers for their helpful feedback.

## Footnotes

[1]Complex RL tasks require similarly complex reward functions, which are often hand-coded. This hand-coding would be very cumbersome for most users, making demonstrations better for training novel behavior.

[2]Although we typically notate functions as uppercase non-bold symbols, we notate the reward function as $\mathbf{r}$, since primarily it will be represented as a parameterized function of the state features and a target for learning.

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
