[Supplementary Material]

# Appendices to
# "Lifelong Inverse Reinforcement Learning"

by **Jorge A. Mendez**, **Shashank Shivkumar**, and **Eric Eaton**

## A   Hessian Derivation for MaxEnt IRL

In this section, we derive the Hessian for the MaxEnt negative log-likelihood in Equation 5 of the main paper. Recall that the Hessian is necessary for updating the $\mathbf{L}$ matrix and computing the task-specific coefficients $\mathbf{s}^{(t)}$. We begin by deriving the log-likelihood:

$$
\begin{aligned}
\log P(\mathcal{Z} \mid \boldsymbol{\theta}) &= \log \prod_{\zeta_j \in \mathcal{Z}} P(\zeta_j \mid \boldsymbol{\theta}, T) \\
&= \sum_{\zeta_j \in \mathcal{Z}} \log \frac{\exp(\boldsymbol{\theta}^\top \mathbf{x}_{\zeta_j}) T_{\zeta_j}}{Z(\boldsymbol{\theta}, T)} \\
&= \sum_{\zeta_j \in \mathcal{Z}} \left\{ \boldsymbol{\theta}^\top \mathbf{x}_{\zeta_j} + \sum_{s_i, a_i, s_{i+1} \in \zeta_j} \log T(s_{i+1} \mid s_i, a_i) - \log Z(\boldsymbol{\theta}, T) \right\} \;,
\end{aligned}
$$

and its gradient:

$$
\begin{aligned}
\nabla_{\boldsymbol{\theta}} \log P(\mathcal{Z} \mid \boldsymbol{\theta}) &= \nabla_{\boldsymbol{\theta}} \sum_{\zeta_j \in \mathcal{Z}} \left\{ \boldsymbol{\theta}^\top \mathbf{x}_{\zeta_j} + \sum_{s_i, a_i, s_{i+1} \in \zeta_j} \log T(s_{i+1} \mid s_i, a_i) - \log Z(\boldsymbol{\theta}, T) \right\} \\
&= \sum_{\zeta_j \in \mathcal{Z}} \mathbf{x}_{\zeta_j} - n \nabla_{\boldsymbol{\theta}} \log Z(\boldsymbol{\theta}, T) \\
&= n\tilde{\mathbf{x}} - n \nabla_{\boldsymbol{\theta}} \log \sum_{\tilde{\zeta}} \exp(\boldsymbol{\theta}^\top \mathbf{x}_{\tilde{\zeta}}) T_{\tilde{\zeta}} \\
&= n\tilde{\mathbf{x}} - n \frac{\sum_{\tilde{\zeta}} \mathbf{x}_{\tilde{\zeta}} \exp(\boldsymbol{\theta}^\top \mathbf{x}_{\tilde{\zeta}}) T_{\tilde{\zeta}}}{\sum_{\tilde{\zeta}} \exp(\boldsymbol{\theta}^\top \mathbf{x}_{\tilde{\zeta}}) T_{\tilde{\zeta}}} \\
&= n\tilde{\mathbf{x}} - n \sum_{\tilde{\zeta}} P(\tilde{\zeta} \mid \boldsymbol{\theta}) \mathbf{x}_{\tilde{\zeta}} \;,
\end{aligned}
$$

where $T_\zeta = \prod_{s_i, a_i, s_{i+1} \in \zeta} T(s_{i+1} \mid s_i, a_i)$ and $\sum_{\tilde{\zeta}}$ indicates $\sum_{\tilde{\zeta} \in \mathcal{Z}_{MDP}}$. Given this gradient, we can now find the Hessian by taking the Jacobian of the gradient as follows:

$$
\begin{aligned}
\nabla_{\boldsymbol{\theta}, \boldsymbol{\theta}} \log P(\mathcal{Z} \mid \boldsymbol{\theta}) &= \mathbf{J}_{\boldsymbol{\theta}} \nabla_{\boldsymbol{\theta}} \log P(\mathcal{Z} \mid \boldsymbol{\theta}) \\
&= n \cancel{\mathbf{J}_{\boldsymbol{\theta}} \tilde{\mathbf{x}}} - n \mathbf{J}_{\boldsymbol{\theta}} \sum_{\tilde{\zeta}} P(\tilde{\zeta} \mid \boldsymbol{\theta}) \mathbf{x}_{\tilde{\zeta}} \\
&= -n \mathbf{J}_{\boldsymbol{\theta}} \frac{\sum_{\tilde{\zeta}} \mathbf{x}_{\tilde{\zeta}} \exp(\boldsymbol{\theta}^\top \mathbf{x}_{\tilde{\zeta}}) T_{\tilde{\zeta}}}{\sum_{\tilde{\zeta}} \exp(\boldsymbol{\theta}^\top \mathbf{x}_{\tilde{\zeta}}) T_{\tilde{\zeta}}} \\
&= -n \frac{\left( \sum_{\tilde{\zeta}} \mathbf{x}_{\tilde{\zeta}} \mathbf{x}_{\tilde{\zeta}}^\top \exp(\boldsymbol{\theta}^\top \mathbf{x}_{\tilde{\zeta}}) T_{\tilde{\zeta}} \right) \cancel{Z(\boldsymbol{\theta}, T)}}{Z^{\cancel{2}}(\boldsymbol{\theta}, T)} \\
&\quad + n \frac{\left( \sum_{\tilde{\zeta}} \mathbf{x}_{\tilde{\zeta}} \exp(\boldsymbol{\theta}^\top \mathbf{x}_{\tilde{\zeta}}) T_{\tilde{\zeta}} \right) \left( \sum_{\tilde{\zeta}} \mathbf{x}_{\tilde{\zeta}}^\top \exp(\boldsymbol{\theta}^\top \mathbf{x}_{\tilde{\zeta}}) T_{\tilde{\zeta}} \right)}{Z^2(\boldsymbol{\theta}, T)} \\
&= -n \sum_{\tilde{\zeta}} \mathbf{x}_{\tilde{\zeta}} \mathbf{x}_{\tilde{\zeta}}^\top P(\tilde{\zeta} \mid \boldsymbol{\theta}) + n \left( \sum_{\tilde{\zeta}} \mathbf{x}_{\tilde{\zeta}} P(\tilde{\zeta} \mid \boldsymbol{\theta}) \right) \left( \sum_{\tilde{\zeta}} \mathbf{x}_{\tilde{\zeta}}^\top P(\tilde{\zeta} \mid \boldsymbol{\theta}) \right) \;.
\end{aligned}
$$

Taking the negative of the final expression and multiplying by $\frac{1}{n}$ as required by our definition, we get the expression in Equation 5 of the main paper.

## B  Additional Experiments Comparing ELIRL to Multi-Task IRL

To analyze how ELIRL performed in comparison to an existing multi-task IRL method, we designed a simplified domain and experimental setting that are compatible with the assumptions behind Babes et al.'s method [2], MTMLIRL. We did attempt to learn the full-size Objectworld using the MTMLIRL code provided by Babes et al. in the BURLAP library [24], but the computational cost of MTMLIRL made it infeasible to run the experiment in a reasonble timeframe. So, in this Appendix, we present results on a simplified Objectworld domain. Even on this simplified domain, we found that MTMLIRL's training time was three orders of magnitude greater than that of ELIRL.

We created 30 Objectworld tasks of size $8 \times 8$, with 3 outer colors and 2 inner colors. We fixed the object locations across tasks to satisfy MTMLIRL's assumption that all tasks use a common state space. Note that the state space of this environment is significantly smaller than that used in experiments described in Section 6.2 of the main paper.

The training procedure for the algorithms was the same as in the main experiments, providing each IRL algorithm with $n_t = 5$ simulated trajectories of length $H = 16$ for each task. We repeated the experiments for 20 different random task orderings for ELIRL and 10 different initializations for MTMLIRL. MTMLIRL was given access to all 30 tasks in batch in order to decrease its computation time while allowing for knowledge sharing across all tasks.

|  | Training time (sec) |
|---|---|
| ELIRL | $1.095 \pm 0.036$ |
| MaxEnt IRL | $1.074 \pm 0.163$ |
| MTMLIRL | $1239.426 \pm 48.373$ |

Figure B.1: Comparison against MTMLIRL. The reward function learned by ELIRL was more accurate than that learned by MTMLIRL on the small-scale Objectworld experiment.

Table B.1: The average learning time per task for ELIRL, MaxEnt IRL, and MTMLIRL on the small-scale Objectworld. The standard error is reported after the $\pm$.

Figure B.1 and Table B.1 show the results of this simplified experiment, which suggest that ELIRL is capable of learning a more accurate reward function than MTMLIRL at a substantially reduced computational cost.

## C  Support for Cross-Domain Transfer

To show how ELIRL can support transfer across tasks from different domains, we adapt ideas from cross-domain transfer in the RL setting, which uses inter-domain mappings. Previous work in this area has used hand-coded mappings [41, 40], learned pairwise mappings autonomously [39, 14, 7, 8, 6, 15, 18, 36], or learned mappings to a shared feature space [5, 3, 23, 22]. Since we consider the case in which there is an unknown (potentially large) number of tasks, we focus upon the last strategy.

In this context, we assume that there is a set of groups $\left\{ \mathcal{G}^{(1)}, \ldots, \mathcal{G}^{(G_{max})} \right\}$ and that all tasks from the same group $t \in \mathcal{G}^{(g)}$ share the feature space $\mathbb{R}^{d_g}$. To transfer knowledge about the reward functions across groups, in addition to learning the shared knowledge base $\mathbf{L} \in \mathbb{R}^{d,k}$, the cross-domain

algorithm learns a set of projection matrices $\boldsymbol{\Psi}^{(g)} \in \mathbb{R}^{d_g,k}$ that specialize the knowledge in $\mathbf{L}$ to each group, such that the parameter vectors are factorized as $\boldsymbol{\theta}^{(t)} = \boldsymbol{\Psi}^{(g)}\mathbf{Ls}^{(t)}$. By modifying the prior in Equation 2 to include a Gaussian prior on the $\boldsymbol{\Psi}^{(g)}$'s and following previous work by obtaining the second-order Taylor approximation [3], we obtain the cross-domain multi-task objective:

$$
\min_{\mathbf{L}} \frac{1}{N} \sum_{g=1}^{G} \min_{\boldsymbol{\Psi}^{(g)}} \sum_{t \in \mathcal{G}^{(g)}} \min_{\mathbf{s}^{(t)}} \left\{ \left( \boldsymbol{\alpha}^{(t)} - \boldsymbol{\Psi}^{(g)}\mathbf{Ls}^{(t)} \right)^{\top} \mathbf{H}^{(t)} \left( \boldsymbol{\alpha}^{(t)} - \boldsymbol{\Psi}^{(g)}\mathbf{Ls}^{(t)} \right) \right.
$$
$$
\left. + \mu \|\mathbf{s}^{(t)}\|_1 \right\} + \gamma \|\boldsymbol{\Psi}^{(g)}\|_{\mathsf{F}}^2 + \lambda \|\mathbf{L}\|_{\mathsf{F}}^2 \ , \tag{C.1}
$$

which can be optimized online via efficient per-task updates. Results showing the effectiveness of cross-domain transfer are presented in Section 6.3 of the main paper.

## D   ELIRL in Continuous State-Action Spaces

To show how ELIRL can easily support other IRL approaches as the base learner, in this section we extend ELIRL to operate in continuous state-action spaces.

When dealing with high-dimensional, continuous environments, it can be prohibitively expensive to discretize the state and action spaces as required in order to use MaxEnt IRL as the base learner for ELIRL. Other existing IRL methods do not require discretization, such as the Relative Entropy IRL algorithm [9] and the Approximated Maximum Entropy (AME) IRL algorithm [20]. The latter of the two techniques is especially suitable to problems in robotics where the demonstrations provided by users are rarely globally optimal, since it assumes only local optimality. We can employ AME-IRL as the single-task learner in ELIRL to make ELIRL compatible with deterministic environments with continuous state and action spaces.

Recall that MaxEnt IRL maximizes the likelihood of the demonstrated trajectories, and that in order to evaluate this equation, we must compute the partition function. This cannot be done in the case of continuous systems as it requires evaluating the policy at every state under a given reward. AME-IRL overcomes this limitation by using the Laplace approximation of the MaxEnt log-likelihood, which models the reward as locally Gaussian, thus removing the dependence on the partition function. The approximate log-likelihood is given by:

$$
\log P(\zeta_j \mid \boldsymbol{\theta}) = \frac{1}{2}\mathbf{g_r}^{\top}\mathbf{H_r}^{-1}\mathbf{g_r} + \frac{1}{2}\log|-\mathbf{H_r}| - \frac{-d_{\mathbf{u}}}{2}\log 2\pi \ , \tag{D.1}
$$

where $\mathbf{g_r}$ and $\mathbf{H_r}$ are the gradient and Hessian of the reward function with respect to the expert user's actions, and $d_{\mathbf{u}}$ is the dimensionality of the actions. In order to optimize the multi-task objective in Equation 4 of the main paper, we also require the Hessian $\mathbf{H}$ of the AME-IRL loss function with respect to the reward parameters $\boldsymbol{\theta}$, which we obtain as:

$$
\mathbf{H} = \frac{\partial \mathbf{h}}{\partial \boldsymbol{\theta}}^{\top} \frac{\partial \mathbf{g_r}}{\partial \boldsymbol{\theta}} + \mathbf{h}^{\top}\frac{\partial^2 \mathbf{g_r}}{\partial \boldsymbol{\theta}^2} - \frac{\partial \mathbf{h}}{\partial \boldsymbol{\theta}}^{\top}\frac{\partial \mathbf{H_r}}{\partial \boldsymbol{\theta}}\mathbf{h} - \frac{1}{2}\mathbf{h}^{\top}\frac{\partial^2 \mathbf{H_r}}{\partial \boldsymbol{\theta}^2}\mathbf{h}
$$
$$
+ \frac{1}{2}\mathrm{tr}\left( \left( \mathbf{H_r}^{-1}\frac{\partial \mathbf{H_r}}{\partial \boldsymbol{\theta}} \right)^2 + \mathbf{H_r}^{-1}\frac{\partial^2 \mathbf{H_r}}{\partial \boldsymbol{\theta}^2} \right) \ , \tag{D.2}
$$

where $\mathbf{h} = \mathbf{H_r}^{-1}\mathbf{g_r}$. This expression can now be plugged into Equation 4 to enable ELIRL to operate in continuous state-action spaces. Section 6.3 of the main paper presents experimental results of applying ELIRL to continuous navigation tasks using AME-IRL as the base learner.