[Reviews · NeurIPS 2018]

Reviewer 1



This paper investigates the problem of inverse reinforcement learning from multiple tasks in a manner that allows transfer of knowledge from some tasks to others. A model using a matrix factorization of task reward functions is presented to realize this. Online methods are then developed that enable this learning to occur in a lifelong manner requiring computation that does not badly scale with the number of tasks. The problem explored is well-motivated in the paper and under-investigated in the existing literature. The paper makes a solid contribution to address this problem. The proposed method adapts the efficient lifelong learning algorithm [33] to maximum entropy inverse reinforcement learning [42]. This seems like a relatively straight-forward combination of ideas, reducing the novelty of contribution. It is unclear what guarantees, if any, are provided by the alternating optimization method along with the approximations introduced by using the Taylor expansion and the sample approximation to compute the Hessian. The experiments show the benefits of the developed methods over reasonable alternative approaches and the authors provide a number of additional experiments in the supplementary materials. My recommendation at this point is “weak accept” since the paper makes a solid technical contribution to the inverse reinforcement learning literature, but without a substantial amount of novelty due to the combination of existing methods or theoretical guarantees. ==== Thank you for addressing my questions in your response.

Reviewer 2



This paper investigates the problem of inverse reinforcement learning in which the agent faces a series of related tasks and has to learn the reward function for each. The framework is fairly straightforward; the paper is well written and easy to read. The proposed framework is based on previous work on life long learning in RL settings with minor incremental changes but still significant. Some additional comments: - intuitively, it looks like this framework will work well when certain aspects of the reward function are shared as in the example of dropping objects. However, what happens when in some tasks an action like dropping an object is bad (e.g., when delivering a tray) while in others it is desirable in some contexts (e.g., dropping an object in the trash bin when cleaning up)? This is not immediately clear and some clarification along with that example would help. - The description of the Objectworld domain could be improved by explicitly stating the goal of the agent in such a task. - A more extensive evaluation with actual human demonstrations would be a big plus. - From the evaluation, it doesn't seem that this framework can scale well for more complicated tasks - if 32 (or 256) trajectories are needed per task and it takes dozens of tasks to converge, would this work in practice where the trajectories come from real people and the tasks are much more complex? If the motivation really is to enable robots to be taught by end-users, it doesn't look like this solution will work well in practice. - Why wasn't the proposed method benchmarked against the method proposed in [11]? - The authors make the case in the related work section that previous methods did not deal with different action spaces; however, it looks like in the experiments all tasks shared the action space. - The conclusion is too short -- no limitations are discussed and little directions for future work

Reviewer 3



Summary: This paper considers the problem of lifelong inverse reinforcement learning, where the goal is to learn a set of reward functions (from demonstrations) that can be applied to a series of tasks. The authors propose to do this by learning and continuously updating a shared latent space of reward components, which are combined with task specific coefficients to reconstruct the reward for a particular task. The derivation of the algorithm basically mirrors the Efficient Lifelong Learning Algorithm (ELLA) (citation [33]). Although ELLA was formulated for supervised learning, variants such as PG-ELLA (not cited in this paper, by Ammar et al. “Online Multi-task Learning for Policy Gradient Methods”) have applied the same derivation procedure to extend the original ELLA algorithm to the reinforcement learning setting. This paper is another extension of ELLA, to the inverse reinforcement learning setting, where instead of sharing policies via a latent space, they are sharing reward functions. Other Comments: How would this approach compare to a method that used imitation learning to reconstruct policies, and then used PG-ELLA for lifelong learning? To clarify, ELIRL can work in different state/action spaces, and therefore also different transition models. However, there is an assumption that the transition model is known. Specifically it would be necessary to have the transition model to solve for pi^{alpha^(t)} when computing the hessian in Equation (5). I liked the evaluation procedure in the experiments, where we can see both the difference between reward functions and the difference between returns from the learned policies. Though I’m not sure what additional information the Highway domain gives compared to the Objectworld. On line 271, k was chosen using domain knowledge. Are there any guidelines for choosing k in arbitrary domains? What was the value of k for these experiments? Please cite which paper GPIRL comes from on line 276. I don’t remember seeing this acronym in the paper (even on line 48). Minor/Typos: In Algorithm 1, I believe the $i$ should be a $j$ (or vice versa)